# Examining the Efficacy of a ‘Feasible’ Nudge Intervention to Increase the Purchase of Vegetables by First Year University Students (17–19 Years of Age) in British Columbia: A Pilot Study

**DOI:** 10.3390/nu11081786

**Published:** 2019-08-02

**Authors:** Matheus Mistura, Nicole Fetterly, Ryan E. Rhodes, Dona Tomlin, Patti-Jean Naylor

**Affiliations:** 1Chronic Disease Prevention Research and Knowledge Exchange Unit, School of Exercise Science, Physical and Health Education, University of Victoria, Victoria, BC V8W 2P1, Canada; 2Behavioural Medicine Lab, School of Exercise Science, Physical and Health Education, University of Victoria, Victoria, BC V8W 2P1, Canada

**Keywords:** nudge, choice architecture, vegetable, food, university, students, adolescents, cafeteria

## Abstract

In the transition from high school to university, vegetable consumption tends to deteriorate, potentially influencing immediate and longer-term health outcomes. Nudges, manipulation of the environment to influence choice, have emerged as important to behavior change goals. This quasi-experimental pilot study examined the impact of a contextually feasible evidence-informed nudge intervention on food purchasing behavior of older adolescents (1st year students) in a university residence cafeteria in British Columbia, Canada. A co-design process with students and staff identified a student relevant and operationally feasible nudge intervention; a placement nudge, fresh vegetables at the hot food table, combined with a sensory and cognitive nudge, signage encouraging vegetable purchase). Using a 12-week single-case A-B-A-B design, observations of the proportion of vegetables purchased were used to assess intervention efficacy. Data analysis included visual trend inspection, central tendency measures, data overlap, variability and latency. Visual trend inspection showed a positive trend when nudges were in place, which was more apparent with female purchases and during the first intervention (B) phase. However, further analysis showed lack of baseline stability, high variability across phases and overlapping data, limiting efficacy conclusions. Menu choices, staff encouragement, term timing and student finances are other potential influences. Further ‘real world’ nudge research is needed.

## 1. Introduction

Vegetables are one of the most important foods we consume as humans due to their nutrient density and fiber, as well as their low energy contribution to the diet. Low consumption levels are linked to poorer health outcomes, including obesity and the development of diseases, such as cardiovascular disease, diabetes and some cancers [1]. The consumption of vegetables can also promote satiety and reduce the risk of obesity, which is a growing concern globally and specifically affects the target population of this study—older adolescents (17–19) in their first year of university, who have documented weight gain [2]. The consumption of vegetables is low in adolescence and tends to deteriorate even more during the transition from adolescence to young adulthood [3]. It has also been found that unhealthy eating habits and weight gain established in youth tend to be maintained during aging [4,5]. This worldwide scenario highlights the importance of developing strategies to increase vegetable consumption in this population and attempt to slow the global obesity epidemic.

Food decisions are made many times throughout the day and are shaped by many factors. The intention to eat well may be modulated by convenience, availability, taste or mood. Environmental interventions that influence these daily food decisions have gained importance as they have been shown to positively shape dietary behaviors [6]. Nudge theory or choice architecture has been gaining momentum in public health to subtly guide an individual’s decisions without denying them autonomy of choice or significantly changing their economic incentives [7]. Nudges have been categorized and shown to be effective with a variety of populations through systematic review and meta-analyses [8,9,10,11]. However, there are not many studies targeting adolescents and evidence of their effectiveness in terms of vegetable intake in the school setting has been labelled as both weak and inconclusive [12].

One categorization scheme emerging from a recent meta-analysis identified three nudge classifications: cognitive/educational, hedonistic/sensory and placement/behavior and found that effect size increased from cognitive to placement/ behavior nudges [9]. An example of a cognitive or educational nudge is the display of nutrition facts at point-of-sale or messaging about the importance of eating vegetables or drinking fewer sugary beverages. A recent Toronto, Canada study found that using physical activity caloric equivalents (PACE) was a more unique and eye-catching way to raise awareness about sugar content in beverages [13].

A sensory nudge might involve descriptive naming of dishes to make them more appealing, for example “farm to table bar” as compared to “salad bar”. This has been tested with adolescents where a cross-country European study targeted adolescents using a sensory nudge (vegetable-forward entrée as the “dish of the day”) and found that the “dish of the day” label did not have a significant effect on vegetable selection compared to the control groups (meat-forward or fish selection) [14,15]. Further, the Danish research group involved in the study did secondary analyses and found that self-reported social norms (e.g., friends and family that were high vegetable consumers) and attitudes had a positive association with choosing the vegetable-forward entrée. In addition, males were much less likely to choose the vegetable-forward dish than females [14].

Finally, a placement or behavior nudge might modify the listing of dishes on a menu to place healthier choices more prominently, or make the healthier choice the default (e.g., side salad not french fries). A placement study conducted with university-aged males in a laboratory setting placed vegetables at the beginning of a buffet versus the end and measured the quantity of vegetables on the plate; finding a significant increase in vegetable selection after exposure to the nudge [16].

Although there are many nudge interventions that show effectiveness, overall there is still a growing need for more real-life studies of scalable nudging strategies targeting adolescents and their food choices. In that context, this pilot study arose with the purpose of implementing a contextually feasible nudge intervention in the University of Victoria’s residence cafeteria to evaluate its impact on vegetable purchases by first-year students (typically aged 17–19 years). A secondary aim was to explore whether this appeared to vary based on the sex of the participant. The hypothesis was that a placement, cognitive and sensory nudge intervention would increase the purchase of vegetables by this population of young adults and this may be more salient among females. UVIC Human Research Ethics Board approved the study (Protocol Number 15-445).

## 2. Materials and Methods

### 2.1. Formative Research and Co-Design Process

Prior to the intervention, along with a review of the literature, formative research was conducted that involved surveying students and conducting focus groups with food services staff to help inform which nudges to trial in this real-life setting with this target audience. This has been referred to in the literature as co-design, allowing for better translation of evidence to practice and resulting in a more successful implementation [17,18]. The web-based student survey asked about their demographics, their daily consumption of vegetables and fruit, and what affects their decision to purchase vegetables. Three hundred and forty adolescent (average age 18) undergraduate students responded to the survey. On average, the adolescents reported consuming 4.6 servings of vegetables and fruit daily (excluding potatoes), approximately half of the recommended amount for their age group. The six reasons most commonly identified for choosing vegetables were (in order of most to least common) healthiness, freshness/appearance, taste/preference, cost, cooking method and convenience (see Table 1). These are similar to other studies with university students [19]. A variety of nudges from the literature that represented the study survey data were presented to food services staff in focus groups to determine which efficacious nudges were feasible for implementation within the operation, and yet addressed what was relevant to the adolescents as per the survey results. Each staff member had three votes to rank their top choices.

Table 2 shows the rankings given by staff on the feasibility of nudges. This data was presented back to the management team, including the dietitian. The co-design of the intervention was thus created jointly by the research team and food services, and the types of nudges were agreed upon by both groups.

### 2.2. Research Design and Sampling

This pilot study utilized a quasi-experimental single case A-B-A-B comparison design. In order to measure the efficacy of the intervention, data was collected for two periods at baseline (A), the first one described the current level of vegetables purchased and the second described whether the behavior continued without the intervention or returned to baseline. The four phases were conducted during the fall term of 2016 over a 10-week period, eliminating both the first few weeks of term when students were adjusting, and the end of term, just prior to the exam period, when cafeteria use changes. Each baseline period lasted two weeks and each intervention phase was three weeks in length. See Table 3 for sample sizes.

### 2.3. Setting

The study was conducted in the main residence cafeteria that serves mostly older adolescent students, on meal plans, who are either first-year or student staff (residence advisors). The cafeteria is not exclusive to this population, so adults that appeared visibly older than the target population of undergraduate students were excluded from the observations. The focus of the nudge was at the “hot table” or “steam line” at lunch and dinner where two choices of entrées are served along with 2–3 choices of side dishes, such as grains, other starches and hot vegetables. An example of an entrée would be lasagna or chicken breasts with mushroom sauce, grain or starches would be items, such as rice, potatoes or noodles depending on the main course, and side dish vegetables could be items such as steamed carrots, pea pods or broccoli. Students make their decisions a la carte and their purchases are charged to their student card in a declining balance.

### 2.4. Intervention

The nudges chosen were placement nudges that involved altering the properties of the vegetables to enhance freshness and appearance. This was implemented by adding an option of fresh, raw vegetables to the existing cooked vegetable option, in combination with an environmental cue (sensory and cognitive) in the form of a small poster displayed at eye level, which highlighted the addition of the fresh vegetable option with a colorful character and message about its health benefits (See Table 4 and Figure 1).

### 2.5. Data Collection

The primary measure was the count of students observed purchasing either one of the vegetable side options (raw or cooked) compared to the total count of students that purchased from the hot table. To address the secondary aim, the counts were recorded by sex (male or female). To provide context for the analysis, prior to each observation period the researcher also recorded all of the foods being served at the hot table and whether they were observing a lunch or dinner. The observations took place during both lunch and dinner for a period of 2 h each from a junction area of about 2.5 m distance with good visibility of the hot table and serving activities. The primary researcher recorded 80% of the observations, with trained assistants supporting the remaining observation periods. To avoid inconsistency, research assistants were provided with a 2-h orientation by the primary researcher on how to observe and track purchases.

### 2.6. Data Analysis

All visual and statistical analyses were made using IBM SPSS V.23^®^ (IBM Corp, Armonk, NY, USA) and Microsoft Excel 2016^®^ software (Redmond, WA, USA). To avoid overestimation of vegetable purchases based on cafeteria attendance, the proportion of vegetables purchased was calculated for each meal daily and then plotted in Excel, both for the overall sample and for females and males separately. Visual inspection is an accepted data analysis technique in single-case multiple baseline designs [20]. Steps in visual inspection as outlined in Kazdin [20] included: superimposing a line representing means for the phases on plots of the daily proportions and examining them visually; and calculating trend lines for each phase and superimposing them onto the data plots to allow for trend analysis where the researcher has looked for changes in the direction or slope of a trend in tandem with the change in study phase (e.g., from A to B).

Additional statistical analyses were conducted to examine mean differences between phases and trends, although, due to ascending and descending trends, these analyses had limitations. The Wilcoxon signed-ranks test was used to determine if the rank of proportions differed significantly across the phases. Descriptive statistics were also generated to allow assessment of variability (which can also be seen visually), latency and overlap between phases. Non-overlapping data refers to points during the baseline that do not reach some or any of the points during the intervention phase [20]. We used above 50%, and ideally >70%, non-overlap as the guideline for concluding efficacy [21].

## 3. Results

Visual inspection of the means used to assess the potential efficacy of the nudge interventions (B1, B2) showed neither intervention had an effect on the mean proportion of vegetables purchased between phases for the overall sample or for females or males analyzed separately. However, with ascending and descending trends, this was not unexpected. The results of the trend analysis are represented in Figure 2, Figure 3 and Figure 4. During the first intervention (B1), there were visible changes in the direction of trend lines compared to baseline (A1) and withdrawal (A2) for both the overall sample and for females and males separately. During the second intervention (B2), the visible differences in trend direction and slope A-B phases were not substantive and control over the outcome variable was not fully demonstrated visually. Further analysis using the Wilcoxon signed-rank test showed that the interventions did not elicit a statistically significant change (see Table 5) in the ranking of the proportion of vegetables purchased from baseline to intervention to withdrawal in either the B1 or B2 phases, and effect sizes were small.

To be considered effective, the percentage of non-overlapping data (PND) must be >50, and ideally >70% [21], and that did not occur in this study. Overall, between the first baseline and intervention phase, the PND was 7.7%, 0% for the re-baseline, and 0% for the re-intervention (all data overlapped). For males only, the first baseline and intervention phase had 0% non-overlapping data. For both males and females, the period between re-baseline and re-intervention had 13.33% PND, and the most substantive amount of non-overlapping data was during the first baseline-intervention phase for females only; equaling 30% PND.

## 4. Discussion

We set out to test the efficacy of contextually relevant nudges targeting older adolescents in their first year of university by conducting a quasi-experiment A-B-A-B design trial in a natural setting: a university residence cafeteria. The evidence-based nudges were selected based on student needs and staff feasibility assessments and were thus potentially more scalable into a ‘real world’ setting if found efficacious. Currently, there is a need for evidence on the impact of nudge strategies on older adolescents, as well as those that have put been put into real world practice and evaluated.

Although visual inspection showed changes in the direction of trends with each presentation and with removal of the intervention, this pattern appeared to weaken over time. Visual inspection also showed high variability in the data, which was confirmed by descriptive measures of variability (standard deviation and range) and by an analysis of overlapping data challenging the strength of conclusions possible from the visual inspection. Although this was also supported by non-significant findings from the Wilcoxon signed-rank tests, the effect sizes were commensurate with the literature [9] and more substantive for females. Combining the multiple analyses conducted in this study failed to definitively support the presence of an intervention effect for the combined placement, cognitive and sensory nudges. These results, although counter to the hypothesis, may be the result of methodological (sample size) and practical limitations, but are also not out of step with other nudge studies, many of which have not shown efficacy [15,23,24,25]. The findings are discussed in the context of the literature below.

Despite student survey results showing that they chose vegetables for health reasons, we know that the university cafeteria and overall food environment decreases students’ intention to make healthy food choices due to the plethora of less healthy foods they must navigate through at every meal opportunity [23]. A couple of research teams have surveyed students and found that health was reported as a primary consideration when choosing foods [19,23], but Bevet and colleagues found that a placement nudge intervention adding vegetable-rich entrées and a healthy snack bar could not steer students away from chicken nuggets [23].

Friis et al. (2017) tested the effects of three nudges, cognitive, placement and perceived nudges, on university-aged students in a lab setting buffet dinner. Only the ‘healthy default’ placement nudge (salad in 200 g jars versus the salad tray in the buffet) resulted in an increase in vegetable intake. The cognitive nudge (benefiting the environment) and perceived variety (different display of salad on the buffet) promoted a reduction in caloric intake but further examination showed that this was due to a lower intake of red meat with no increase in vegetable consumption [24].

A previous study that tested nudges with European adolescents showed that a salience nudge was not effective and highlighted the importance of other psychosocial factors, such as social norms, sex and attitudes [14]. A systematic review of interventions targeted at university-aged students showed a broader set of behavior change techniques that were effective, including nutrition education and enhanced self-regulation, as well as point-of-choice messaging (which can be classified as a cognitive or salience nudge) were effective options [26]. Sunstein (2017) discussed the reasons that nudges fail, one being strong antecedent preferences of the target [27].

A non-randomized intervention conducted by Broers and colleagues (2019) showed that cues increased selections from the hot vegetable buffet in a university cafeteria, but increasing the accessibility of pre-biotic vegetables and cues (tray liners promoting prebiotic vegetables) decreased the chances of students picking the target vegetable [28]. The variability in findings from previous nudge studies has been demonstrated in systematic reviews [9,10,11].

Novel attempts to educate consumers at the point of selection may be needed, rather than straight calorie labeling or cognitive messaging identifying foods as ‘healthy’. Scourboutakos et al. (2017) stated that young people want more shocking educational messaging. They found an effect in their real-life university cafeteria study that used physical activity caloric equivalents (e.g., minutes exercising to burn the calories from the beverages) labeling to reduce sugary beverage purchases, despite it being an all-you-care-to-eat cafeteria where cost was not a factor in selection [13].

### Strengths and Limitations

The study is one of the few nudge interventions applied in a real-world cafeteria setting and targeting adolescents. Identifying the feasibility and efficacy of scalable nudges is important for public health decision-making. The formative work to ensure the evidence-based nudges were contextually relevant to students and staff is a strength, as was the contextually relevant research design, notwithstanding the limitations described below. Additionally, numerous analytical approaches were adopted to ensure a transparent and fulsome analysis, and effect sizes commensurate with previous literature [9] were identified. Conversely, there were many limitations, which is a result of the real-life context and related research design. First, the single case A-B-A-B research design is a quasi-experimental design and relies on a return to baseline or change in the direction or slope of a trend after removal of the intervention, followed by a replication of the effect following re-presentation of the intervention to demonstrate ‘control’ over the effect [20]. The trend weakened over time, bringing into question whether there was an intervention effect. There appeared to be a replication of the effect, but the trend weakened over time, challenging the conclusion that there was experimental control. As discussed earlier there was high variability in the data (overlapping data was high) and data stability is also a pre-requisite to demonstrating an effect in single case designs [20].

The research design did not include a comparison group, nor were adjustments for confounding variables possible. The observation itself may have acted as a salience nudge. Furthermore, although data was collected for 4 h/day across lunch and dinner for a full university term, and a large number of meals were observed (>5000 meals per phase), it was a relatively small data set in the analysis with only one data point per day (the proportion of vegetable servings per hot table purchases).

The study would have had to have approximately 100 data points per phase to detect the effect sizes achieved between baseline and intervention phases if a parametric design was used. There were only 186 days in the university year, and implementation over the entire period was pragmatically impossible. The small to modest effect sizes between the first baseline and intervention and the second baseline were comparable to typical nudge effect sizes reported by Cadario and Chandon in 2017 [9], suggesting that a more robust sample size may have increased the likelihood of seeing a significant effect.

The intervention seemed to visibly lose efficacy over time, with minimal changes in trends in the second phase compared to the first phase. The nudges may have lost their impact due to students’ familiarity with the addition of the fresh vegetables, lack of variety in the fresh vegetable offering (observed by the research assistant) and/or elements related to the signage and vegetable tray placement. For instance, the primary offering was raw carrot and celery sticks, which may be considered lower value or ‘conventional’ vegetables. Sunstein named five reasons why nudges may fail, and having a short-term effect was one of them, as well as strong antecedent preferences, perhaps including the preferred type of vegetables [27]. The choice of raw vegetables was out of the control of the research team and was determined by the affordability and ease of preparation by food service staff. In a future intervention, using other fresh vegetables, may increase the selection of vegetables by students [28,29].

The signage or cognitive messaging used was also limited by time and cost of designing and printing. Floor decals and large posters may have proved more effective and eye-catching. Naming the vegetables (a greater sensory nudge) and the variety and placement of the nudges may also have improved the effect [30].

Additionally, pragmatic considerations (e.g., distance from the steam unit) led cafeteria staff to place the raw vegetable trays at the end of the food offerings rather than the beginning. This may have influenced the results as previous research has shown that food positioning affects food choice [16,28]. Kongsbak et al. found that placement of vegetables in different bowls at the beginning of a buffet significantly increased selection as compared to the vegetables served altogether in one bowl and at the end of the buffet after the pasta and bread choices. This was conducted in a laboratory, not a real-world setting, and only once, so the results may have varied if issues like price and habituation were encountered, which likely occurred in this real-life study [16,27].

Other factors were elucidated by cafeteria staff anecdotally and included the financial situation of students over the term. Budgetary concerns may have resulted in the purchase of similarly priced highly satiating foods (e.g., fried potatoes, perogies) over less satiating vegetables [29]. Stressful events such as exams, projects, and the end of the term may have also influenced the students’ choice for more ‘comforting’ foods [31]. Control over what else was available in the cafeteria (e.g., grill items, french fries) also belonged solely to the food service staff and not the researchers. That said, nudges are supposed to work without the limitation of choice to the consumer.

Another issue that developed in this setting was menu composition, which was out of the control of the research team. Specifically, the menu composition was two entrées, usually animal proteins, two starchy side dishes, one option of cooked vegetables, and during the intervention period, the raw vegetables (crudités). Often the color of the entrée was similar to the vegetable offering (observation by research staff). Also, each item was purchased a la carte at $6.95 for each entrée and $2 for each option of side dishes and vegetables. Purchasing each item separately has been shown to decrease the purchase of vegetables, while a combination of products with a fixed single price (such as an entrée plus vegetable) increases the purchase of vegetables [32,33,34]. Finally, the perceived value assigned to certain products may have been problematic. It is possible that students in this study perceived that $2 for a portion of vegetables was not good value and would rather spend their meal plan dollars on other, more calorically-dense items (e.g., entrées, fries, drinks, desserts) [29].

Overall, there are many more unintended nudges occurring in a cafeteria than the one being studied, for example the smell of fries, the colorful packaging of soda pop and the ordering and naming of items on a menu. Although the co-design methodology did result in staff buy-in and successful implementation, it did not prove successful in increasing adolescent student vegetable consumption despite the engagement of the target audience and their needs via the student survey. Sunstein (2017) cautioned that sometimes a “plausible (and abstractly correct) understanding about what drives human behavior turns out to be wrong in a certain context”, p. 5 [27].

## 5. Conclusions

Co-design of interventions in real-life settings is related to better implementation, as interventions typically reflect and/or are relevant for the context [17,18]. The impact of this intervention on older adolescent vegetable purchasing remains in question but the effect sizes were promising. Thus future research should incorporate innovative research designs and achieve power in such real world contexts. Nudge researchers must also accommodate for other operational issues within the full cafeteria that may over-power the effect of small nudges in constrained areas. More research is needed in this field and specifically in this target population, where there is lower than recommended vegetable consumption and documented weight gain. Nudges in real-life settings, rather than laboratories, need to have more controls in place that also balance the operational needs of the food service establishment. They may also need to be strengthened or speak to the ‘actual’ issues of the target population using more salient cognitive messages, innovative use of social media and reflect current food trends. Interventions may need to be incentivized to mitigate decreases in consumption and establish healthy eating habits throughout the lifespan [26]. These findings will inform re-design and testing of future nudges. in real world settings.

## Figures and Tables

**Figure 1 nutrients-11-01786-f001:**
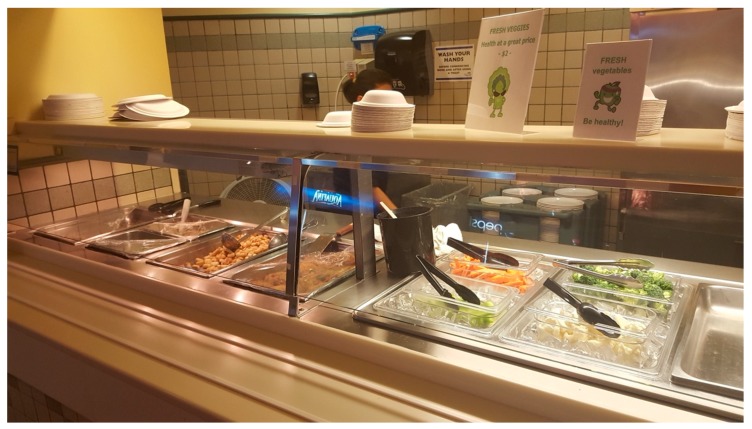
Display of sensory and cognitive nudges and hot table placement of fresh vegetables.

**Figure 2 nutrients-11-01786-f002:**
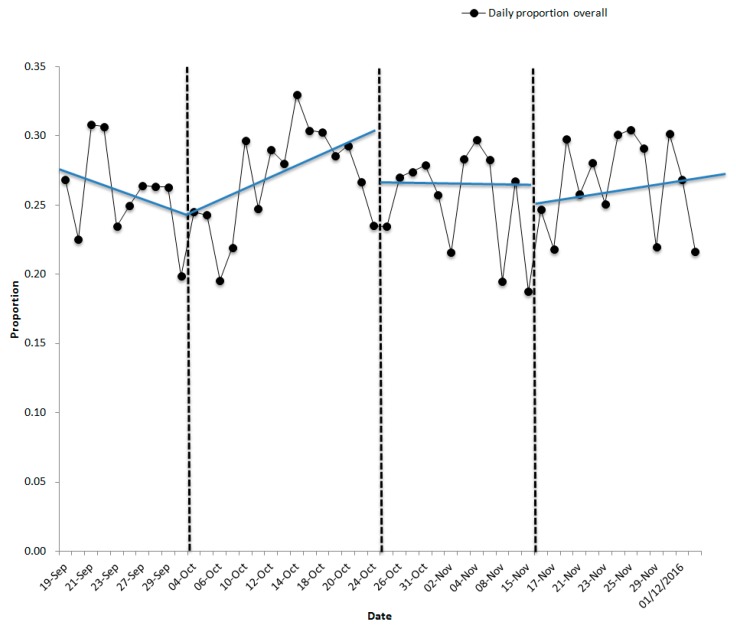
Proportion of vegetables purchased and trend lines across all periods for the overall sample in the university cafeteria.

**Figure 3 nutrients-11-01786-f003:**
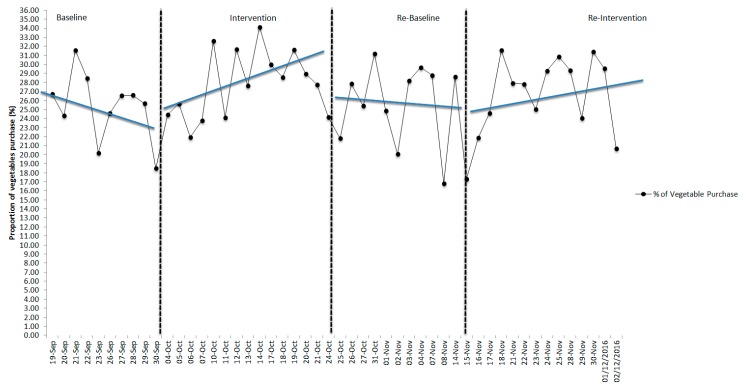
Proportion of vegetables purchased and trend lines across all periods for the female young adults in the university cafeteria.

**Figure 4 nutrients-11-01786-f004:**
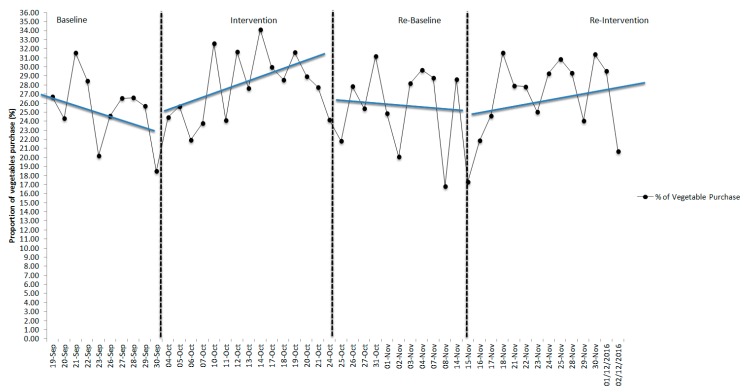
Proportion of vegetables purchased and trend lines across all periods for the male young adults in the university cafeteria.

**Table 1 nutrients-11-01786-t001:** Qualitative summary of the main drivers of vegetable purchase by adolescents.

Top Reasons	% of Students Citing the Reason as Important	Main Themes
Healthiness	36%	Healthy; vitamins; nutrients; feel good; good for you
Freshness	11%	Freshness; look fresh
Taste	10%	Taste; personal preference
Cost	9%	Cost; price
Cooking method	5%	Steamed vs fried; cooked; frozen; raw; added sauce
Convenience	4%	Quick; time; easy access

**Table 2 nutrients-11-01786-t002:** Ranking of nudge feasibility by the staff focus group (n = 8).

Nudge	Votes
Vegetables as default	8
Convenience ‘to go’	7
Increased choice	6
Veg 1st on menu	3
Taste testing, fresh veg side	2
Enhance appearance	1
Color coded sign	0
Pathway	0

**Table 3 nutrients-11-01786-t003:** Number of data points and number of hot table purchases by sex and phase.

	Baseline 1	Intervention 1	Baseline 2	Intervention 2
Data points per phase	n = 10	n = 13	n = 12	n = 15
Hot Table Purchases	n	n	n	n
Females	2037	2566	2476	3063
Males	3061	3607	3278	4322
Overall	5098	6173	5754	7385

**Table 4 nutrients-11-01786-t004:** Nudge interventions by category.

Nudge Categorizations	Study Interventions
Placement Nudge	Adding raw vegetables on the hot line, not just at the salad bar, making them easier to choose with an entrée.
Hedonistic/Sensory	Adding colourful, fresh vegetable option alongside cooked; colourful poster (see Appendix A).
Cognitive/Educational	Poster with messaging about vegetable benefits (see Appendix A).

**Table 5 nutrients-11-01786-t005:** Analysis of the Wilcoxon signed-rank test across three phases with means.

Condition	A1–B1 Means (SD) Statistic, *p* Effect Size *	B1–A2 Means (SD) Statistic, *p* Effect Size *	A2–B2 Means (SD) Statistic, *p* Effect Size *
Overall	$\bar{X}$^1^ = 25.81 (3.38) $\bar{X}$^2^ = 26.51 (4.17) Z = −0.459, *p* = 0.646 D = 0.192	$\bar{X}$^2^ = 26.51 (4.17) $\bar{X}$^3^ = 26.28 (2.63) Z = −0.561, *p* = 0.575 D = 0.226	$\bar{X}$^3^ = 26.28 (2.63) $\bar{X}$^4^ = 26.13 (3.77) Z = −0.051, *p* = 0.959 D = 0.02
Female	$\bar{X}$^1^ = 25.30 (3.78) $\bar{X}$^2^ = 28.05 (3.70) Z = −0.968, *p* = 0.333 D = 0.412	$\bar{X}$^2^ = 28.05 (3.70) $\bar{X}$^3^ = 25.33 (4.4) Z = −0.746, *p* = 0.445 D = 0.302	$\bar{X}$^3^ = 25.33 (4.4) $\bar{X}$^4^ = 26.65 (4.25) Z = −0.051, *p* = 0.959 D = 0.02
Male	$\bar{X}$^1^ = 26.24 (3.60) $\bar{X}$^2^ = 26.80 (4.02) Z = −0.255, *p* = 0.799 D = 0.106	$\bar{X}$^2^ = 26.80 (4.02) $\bar{X}$^3^ = 26.40 (2.81) Z = −0.153, *p* = 0.878 r = 0.060	$\bar{X}$^3^ = 26.40 (2.81) $\bar{X}$^4^ = 26.40 (3.68) Z = −0.051, *p* = 0.959 r = 0.010

* Effect sizes estimated as Cohen’s D using https://www.psychometrica.de/effect_size.html [22].

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
