# Peer review of "Examining the Efficacy of a ‘Feasible’ Nudge Intervention to Increase the Purchase of Vegetables by First Year University Students (17–19 Years of Age) in British Columbia: A Pilot Study"

_nutrients, 2019, doi:10.3390/nu11081786_

Round 1
Reviewer 1 Report
The authors provide some examples and conceptual definitions of nudges from the literature in their background. It is not clear to me how they operationalized these constructs in their own study. In the intervention section of the methods, the first line mentions the use of a placement nudge and then it gets confusing when the authors mention that the placement nudge was combined with sensory and cognitive cues. A table could be provided with what types of nudges were used and how these were operationalized in the present study with appropriate references.
The intervention description in the methods is weak. It needs to be grounded within the Nudge Theory mentioned in the background and connected to the study.
It is also not clear why students purchasing raw/cooked options were compared to those buying the hot options. Does that mean that there were no vegetables included in the hot meals provided?
While the primary measure is mentioned in the data collection section, the authors left out details of what other variables they measured.
Why did the authors choose to compare males and females?
There is no mention of the sample size in the methods or in the results, except for the focus groups. It would be good to provide the number of participants during each phase.
Discussion
Nudges are similar to cues to action mentioned in several health behavior theories such as the Health Belief Model. The authors could look as to why cues to action are not always an effective way to change behavior which in this case is vegetable consumption. Individuals' behavior can be changed by altering the environment or by changing their immediate beliefs underlying the behavior. In this study, the researchers worked on changing the environment primarily to influence healthier eating choices.
Some other limitations which the authors did not include were the lack of a control group and confounding factors were not controlled in the analyses.
Reviewer 2 Report
The idea of the nudging intervention for this paper seems to be good, but the paper needs some improvements as stated below:
Introduction
Some key references on nudging strategies towards adolescents (especially in a real foodservice setting and systematic reviews) are missing, for example:
dos Santos, Q., Perez-Cueto, F.J.A., Rodrigues, V.M. et al. Impact of a nudging intervention and factors associated with vegetable dish choice among European adolescents
Eur J Nutr (2019). https://doi.org/10.1007/s00394-019-01903-y
dos Santos, Q, Nogueira, BM, Rodrigues, VM, et al. Nudging using the ‘dish of the day’ strategy does not work for plant‐based meals in a Danish sample of adolescent and older people. Int J Consum Stud. 2018; 42: 327– 334. https://doi.org/10.1111/ijcs.12421
Nørnberg, T. R., Houlby, L., Skov, L. R., & Peréz-Cueto, F. J. A. (2016). Choice architecture interventions for increased vegetable intake and behaviour change in a school setting: a systematic review. Perspectives in Public Health, 136(3), 132–142. https://doi.org/10.1177/1757913915596017
Methods
I did not really understand, has the paper presented data only from the pilot study? While I was reading the paper, I thought it would present data from the pilot study followed by data from the “real study”. But it seems that only data on pilot study is presented. The authors should emphasize it more in the abstract.
Have you calculated the sample size needed for the study? If yes, why you did not state it?
Page 4, lines 109-111: ” The focus of the nudge was at the “hot table” or “steam line” at lunch and dinner where 2 choices of entrees are served along with 2-3 choices of side dishes like grains, other starches and hot vegetables.” It is not clear, what kind of entrees were usually served before/during the intervention? What kind of side dishes? Provide examples.
Lines 117-118: “in the form of a small poster displayed at eye level highlighting the addition of the fresh 118 vegetable option with a colourful character and messaging about good value for health benefits”-. Have you considered to add a figure with an example of one of the posters used? It would be a good idea to illustrate this strategy.
Lines 120-121: “The primary measure was the count of students observed purchasing either one of the vegetable options (raw or cooked) compared to the count of students that purchased from the hot table.” Have you considered tin the analysis, he two options of vegetables together (raw and cooked) versus the hot table? Or separately? And again, provide some examples of the dishes normally served in the hot table.
Lines 131-132: “Visual inspection is an accepted data analysis technique in single-case multiple baseline designs [15].” Could you add more details on this technique, since the reference is a book and not everyone has access to it?
Results
The number of participants included in the study was not showed. As well as sociodemographic characteristics, such as mean age, % female, among others.
Lines 151-154: “Further analysis using the Wilcoxon signed-rank test showed that the interventions did not elicit a statistically significant change (see Table 3) in the ranking of the proportion of vegetables purchased from baseline to intervention to withdrawal in either the B1 or B2 phases.“ Again, if you have calculated the needed sample size to find a difference in the results, you have to state this information in the text. In addition, you should include the information on the number of participants of the study to make a comparison between targeted sample size versus real sample size.
Discussion
Lines 183-184- Other important references on the topic are missing, as previously stated.
Lines 209-210: “The intervention seemed to visibly lose efficacy over time, with minimal changes in trends in the second phase compared to during the first phase.” This was also found in previous studies according to the literature, I suggest including those references here.
One question is: Have the researchers considered that time for intervention (number of weeks) enough when compared to similar previous studies? And the time in between interventions? I would like to hear more about that.
To build a stronger discussion, the authors should have added details in relation to the number of participants and % males and females, calculated sample size versus real sample size, the findings in comparison to key studies already published and so on. I think this paper needs to be improved in general.
Round 2
Reviewer 1 Report
Thank you for improving the paper based on the comments we provided.
Author Response
Based on the comment that the reviewer was satisfied with the changes we made no further revisions.
Reviewer 2 Report
This paper has too many limitations: there was high variability in the data, the research design didn’t include a comparison group nor were adjustments for confounding variables possible. Furthermore it was a relatively small data set with only one data point per day (the proportion of vegetable servings per hot table purchases). The study would have had to have approximately 100 data points per phase to detect the effect sizes achieved between baseline and intervention phases if a parametric design was used. This means that the study was underpowered, and essential steps such as calculation of sample size needed to find a significant effect prior to the experiment is lacking.
